# Label-free 3D molecular imaging of living tissues using Raman spectral projection tomography

Elzbieta Stepula [1,6], Anders R. Walther[2,6], Magnus Jensen[1], Dev R. Mehrotra[3], Mu H. Yuan[1], Simon V. Pedersen[4], Vishal Kumar [1], Eileen Gentleman [1,5], Michael B. Albro[3], Martin A. B. Hedegaard [2] ✉ & Mads S. Bergholt [1] ✉

The ability to image tissues in three dimensions (3D) with label-free molecular contrast at the mesoscale would be a valuable capability in biology and biomedicine. Here, we introduce Raman spectral projection tomography (RSPT) for volumetric molecular imaging with optical sub-millimeter spatial resolution. We have developed a RSPT imaging instrument capable of providing 3D molecular contrast in transparent and semi-transparent samples. We also created a computational pipeline for multivariate reconstruction to extract label-free spatial molecular information from Raman projection data. Using these tools, we demonstrate imaging and visualization of phantoms of various complex shapes with label-free molecular contrast. Finally, we apply RSPT as a tool for imaging of molecular gradients and extracellular matrix heterogeneities in fixed and living tissue-engineered constructs and explanted native cartilage tissues. We show that there exists a favorable balance wherein employing Raman spectroscopy, with its advantages in live cell imaging and label-free molecular contrast, outweighs the reduction in imaging resolution and blurring caused by diffuse photon propagation. Thus, RSPT imaging opens new possibilities for label-free molecular monitoring of tissues.

Label-free imaging of tissues at the mesoscale in three dimensions is of substantial importance across biomedical sciences. Mesoscale volumetric imaging techniques, characterized by intermediate spatial resolutions spanning from hundreds of micrometers to millimeters and centimeters provide a comprehensive view of complex biological systems, allowing researchers to investigate relationships between cellular structures, tissue organization, and organ function. This comprehensive insight into the intermediate scale is essential for deciphering the multi-scale nature of tissues across various levels of organization.

Numerous three-dimensional (3D) imaging techniques have been applied for imaging at the mesoscale. Computed tomography (CT) and micro-CT use X-ray absorption while magnetic resonance imaging (MRI) employs strong magnetic fields and radio waves to generate 3D volumetric images of tissue structures[1,2]. Optical methods for mesoscale imaging include optical projection tomography (OPT)[3], photoacoustic tomography[4], and diffuse optical tomography (DOT)[5]. OPT is used widely in biomedical research with applications that include developmental biology, cancer research, and neurosciences[6–8]. OPT measures optical projections of the emission/absorption of light to generate volumetric images of the structure of transparent or optically cleared biological samples[6]. While these techniques can provide volumetric spatial information at various scales, they lack the ability to directly depict the molecular composition of the specimen without labeling.

[1]Centre for Craniofacial & Regenerative Biology, King's College London, London, UK. [2]SDU Chemical Engineering, Faculty of Engineering, University of Southern Denmark, Odense, Denmark. [3]Department of Mechanical Engineering, Boston University, Boston, MA, USA. [4]SDU Biotechnology, Faculty of Engineering, University of Southern Denmark, Odense, Denmark. [5]Department of Biomedical Sciences, University of Lausanne, Lausanne, Switzerland. [6]These authors contributed equally: Elzbieta Stepula, Anders R. Walther. ✉e-mail: marhe@igt.sdu.dk; mads.bergholt@kcl.ac.uk

Imaging at a micrometer scale relies heavily on microscopy techniques such as confocal laser-scanning microscopy and light sheet microscopy. These platforms enable high-resolution volumetric imaging of labeled structures, such as live organoids or zebrafish models, etc.[9–11] Multiphoton imaging approaches such as second harmonic generation, two-photon excited fluorescence, coherent anti-stokes Raman spectroscopy and stimulated Raman spectroscopy (SRS) offers rapid label-free volumetric molecular imaging. However, these techniques are limited by a small field of view (FOV) and imaging depths due to the requirement of high numerical aperture objectives[12–16].

Raman spectroscopy probes the specific molecular vibrations of the molecules, thus offering comprehensive information about the structure and composition of biological tissue. Confocal Raman microspectroscopy has long been used to provide 3D images of the chemical composition with diffraction-limited resolution and has been used for imaging cells and organoids. However, Raman microspectroscopy is limited to submillimeter tissue penetration depths[17–20]. For molecular analysis in deeper tissues (from a few microns to several millimeters or centimeters), special techniques such as spatially offset Raman spectroscopy (SORS)[21–23] have been developed. However, in the context of imaging, the major disadvantage of SORS is its limited spatial information due to the collection of diffusely scattered photons. The scattering of photons in tissue, which results in diffuse propagation, is frequently described by metrics such as the transport mean free path (TMFP). The scattering phenomenon in tissue poses a challenge to achieving high-fidelity optical projections. Consequently, diffuse Raman tomography has been developed which offers label-free volumetric analysis in 3D[24–26]. In diffuse Raman tomography, reconstructing images requires inverse modeling. Therefore, merely collecting Raman spectra from various angles around the sample is insufficient to generate a 3D image. Instead, finite element methods are employed to reconstruct the spatial distribution of Raman-active molecules within the sample volume[24–26]. Diffuse Raman tomography is therefore also often combined with other imaging modalities such as micro-CT[27,28]. Another recent development in projection tomography involves using a bandpass filter to capture isolated Raman scattering bands using visible excitation[29]. However, this approach falls short in utilizing the comprehensive molecular information present in full Raman spectra, might be obscured by autofluorescence, and has not been demonstrated to be compatible with applications involving living tissues. Various methods for volumetric imaging utilizing SRS have also been developed, including stimulated Raman projection tomography[16,30]. These methodologies are primarily applicable to microscopic-sized samples.

A recent development, pulse-sheet chemical tomography[31], offers a new approach for imaging millimeter-sized samples with molecular contrast, however, with limited chemical specificity owing to its spectral resolution of 160 cm$^{-1}$. Advanced bond-selective diffraction tomography employing photothermal infrared techniques has also been demonstrated to enable rapid (~20 s) tomographic reconstruction with high resolution, albeit within a microscopic FOV[32]. Considering the strengths and limitations of current techniques, there remains a clear demand for the advancement of an optical imaging method that combines high molecular specificity, a broad mesoscale FOV, and the capacity to image living tissues without causing damage.

In this work we introduce Raman spectral projection tomography (RSPT) for volumetric label-free molecular spectral imaging at mesoscale. We developed a transmission Raman spectral projection tomograph and established a comprehensive data analysis framework to extract highly specific molecular information from projected Raman spectra. First, we demonstrate accurate label-free 3D molecular imaging of phantoms of various complex shapes supported by ray tracing simulations. We show that there exists a favorable balance wherein employing Raman spectroscopy, with its advantages in live cell imaging and label-free molecular contrast, outweighs the reduction in

imaging resolution and blurring caused by diffuse photon propagation. We demonstrate the feasibility of RSPT as a tool for evaluating the spatial heterogeneities of extracellular matrix (ECM) in soft connective tissue specimens of native and engineered articular cartilage. Finally, we show that the developed method can be applied to the imaging of living cells/tissues, highlighting the capability of the tool for temporal non-destructive monitoring of tissue-engineered (TE) constructs in their critical early phases of growth.

## Results

### Development of RSPT

Constructing a Raman spectral projection tomographic imaging instrument demands meeting stringent requirements similar to those essential for OPT for image formation but entails additional challenges specific to forming images based on spectral data. Diffuse light scattering is highly detrimental to any image formation in projection tomography. Laser excitation light and Raman scattered photons undergo multiple scattering events as they propagate through tissue, resulting in the breakdown of the parallel beam assumption with resulting blurring in the reconstructed image. Consequently, to form the best possible projection images, it is essential to capture the forward scattered Raman photons while rejecting diffuse light contributions from the sample. Secondly, traditional OPT captures full 2D images at each angular projection while it would require a new optical design to facilitate the acquisition of full-spectrum Raman projections. Thirdly, the optical projections in a Raman spectroscopic setting would consist of compound Raman spectra accumulated through the sample, necessitating new sophisticated data preprocessing and analysis techniques to be developed.

We have developed an RSPT instrument based on a telecentric transmission Raman imaging geometry. By rotating and translating the sample along the z-axis, 2D projections can be measured from all angles (Fig. 1A). In this design we excite the sample and collect a full z-slice of Raman spectra covering the fingerprint (600–1800 cm$^{-1}$) and high-wavenumber range (2700–3600 cm$^{-1}$) to offer comprehensive molecular information[33]. Measuring a single slice at a time reduces the collection of out-of-plane diffuse Raman photon contributions. The transmission Raman approach provides benefits compared to a reflective configuration, since transmission Raman allows for the accumulation of photons through the sample, as opposed to mostly capturing Raman signals near the surface. We integrated a custom-manufactured high-power (2.0 W) near-infrared (NIR) 785 nm laser for excitation allowing sufficient Raman signals to be generated within bulk ~1 cm semi-transparent tissues. The incident laser beam was shaped into a line (power density of ~20 W/cm$^2$) using a long focal length cylindrical lens, which illuminated the sample similarly to slice-illuminated OPT[34]. The focal plane was situated at the midpoint of the rotation axis. Each projection contains Raman photons from the entire z-slice although the laser light is attenuated through the sample. To reduce the collection of the diffusely scattered Raman light, we employed a pair of long focal length and low NA imaging lenses with a stop aperture in a telecentric design that collects the Raman scattered photons propagating parallel through the sample. The collected Raman photons were then projected onto a custom-manufactured line-to-line fiber bundle consisting of 43 cores (50 µm) acting as a slit. The line-to-line fiber array was coupled to a high-throughput spectrometer and imaged onto a 2000 × 256-pixel charge-coupled device (CCD) for a single z-slice spatial acquisition of Raman spectra (Fig. 1A). Following the binning of pixels covering individual fibers on the CCD, this configuration provides snapshots of 43 Raman spectra across the FOV. In this way, a full projection dataset can be obtained by rotation and z-translation of the sample. To accompany the instrument, we have developed comprehensive controlling software for the fully automated data acquisition and analysis (Fig. S1).

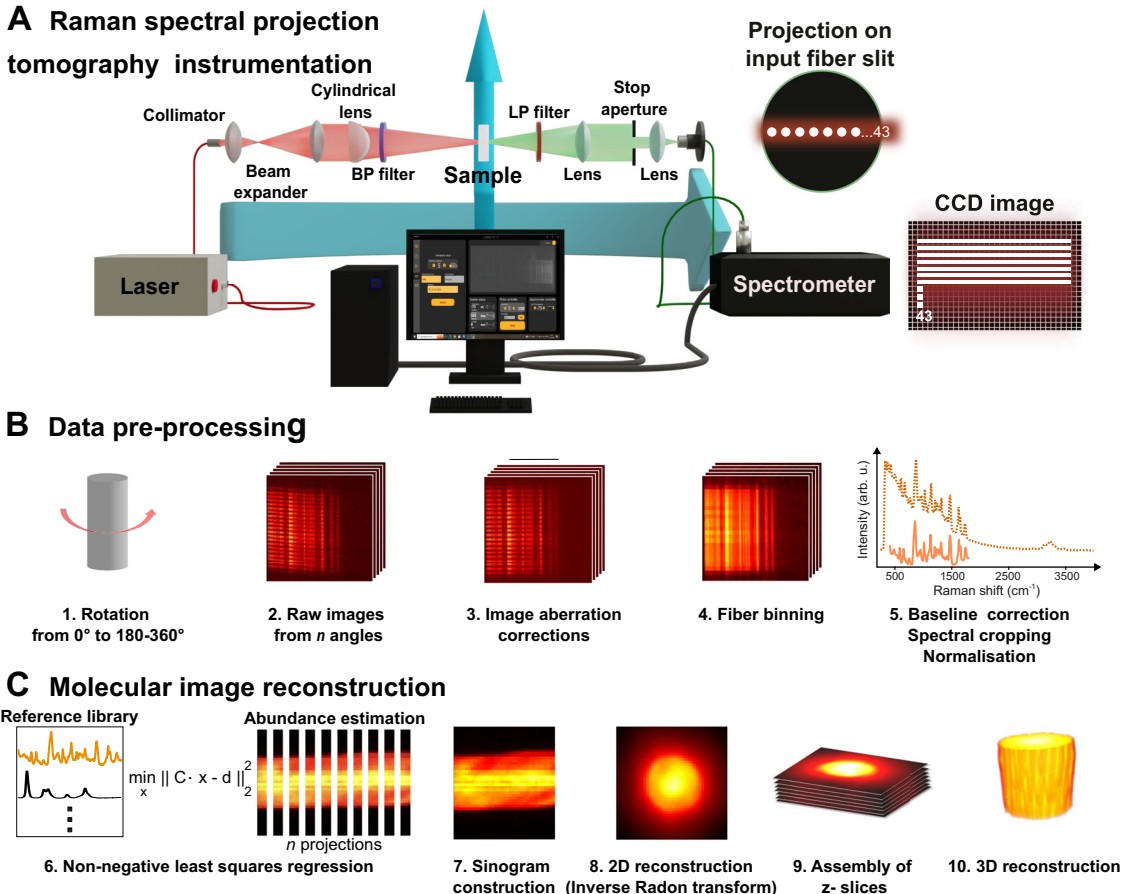

**A Raman spectral projection tomography instrumentation**

**B Data pre-processing**

1. Rotation from 0° to 180-360°
2. Raw images from *n* angles
3. Image aberration corrections
4. Fiber binning
5. Baseline correction Spectral cropping Normalisation

**C Molecular image reconstruction**

6. Non-negative least squares regression
7. Sinogram construction
8. 2D reconstruction (Inverse Radon transform)
9. Assembly of z-slices
10. 3D reconstruction

**Fig. 1 | RSPT instrumentation and analysis. A** Schematic illustration of the RSPT imaging instrument: a 2 W 785 nm laser is beam-shaped into a line using a cylindrical lens. The transmitted Raman scattered light from the sample is collected and imaged onto a line array comprised of 43 fiber cores coupled to a spectrometer. The Raman signals are then captured with a 2D CCD. The sample can be rotated up to 360˚ and displaced along the *z*-axis to collect a full 3D projection dataset. **B** Depiction of data pre-processing workflow: pre-processing involves computational correction of image aberrations in raw CCD images, binning pixels containing individual fibers, and standard Raman processing including smoothing, baseline correction, and cropping to the spectral fingerprint (800–1800 cm⁻¹) and high wavenumber region (2700–3600 cm⁻¹). Finally, spectra are vector normalized. **C** The image reconstruction phase employs multivariate non-negative least squares regression analysis using a library of pure component Raman spectra. Sinograms are constructed and a 2D reconstruction is performed using the inverse radon transformation. Sequential application to all *z*-slices enables the 3D reconstruction for volumetric rendering.

Each projected plane generates a vast amount of Raman data along the height of the CCD (256 Raman spectra per projection angle) that are distorted due to spectrometer aberrations. This necessitates CCD image correction, spectral pre-processing, multivariate analysis, 2D reconstruction and 3D rendering, which poses a significant computational challenge. We have developed a data analysis pipeline incorporating data pre-processing including spectrometer image aberration correction for each fiber to ensure high spectral resolution, horizontal binning of fiber projections, wavelength calibration, autofluorescence subtraction, and normalization (Fig. 1B, see "Methods"). To enable molecular imaging and reconstruction, we employ multivariate regression of a library of purified reference components (Fig. 1C), as well as a background spectrum[35]. In this approach, we measure Raman spectra of purified basis constituents and utilize non-negative least squares regression analysis to quantify the relative abundance of each constituent. This implies that the composite Raman spectra observed result from the accumulation of all Raman signals generated within the sample. For highly transparent samples, considering that only a negligible portion of the incident laser light undergoes Raman scattering, the laser intensity remains mostly unchanged through samples, thus supporting the general assumption for image formation in projection tomography. Finally, 3D image reconstruction is performed using inverse Radon transform−back projection of the regression component

abundances for each slice (Fig. 1C)[36]. The molecular abundances within the resulting voxels are ultimately normalized to one, facilitating the extraction of relative molecular gradients in samples. This implies that laser attenuation will have less effect on the reconstruction.

## Benchmarking of RSPT imaging performance

RSPT imaging is flexible and allows for the use of various magnifications, numerical apertures and, consequently, different spatial resolutions by modifying the collection imaging optics. Here we benchmarked the RSPT with an FOV of ~9 mm (Fig. 2A) and a magnification factor of ~0.4 (voxel size of $0.2 \times 0.2 \times 0.2$ mm), which were used for further experimental imaging in this work. To characterize the system's optical contrast, we 3D printed and fabricated three groups of spatial resolution targets with aperture widths of 1000 μm, 500 μm, and 400 μm, using transparent resin. These targets were placed and imaged at a 45-degree angle to the optical axis (Fig. 2B). This diagonal arrangement allowed us to assess the system's capacity to resolve details at varying depths within the sample in 2D. From each sample, we collected Raman projections and analyzed the $CH_2$ stretching vibration of the resin polymer (2940 cm⁻¹) along the depth of field (Fig. 2C)[37]. We then determined the contrast ratio defined as the intensity difference between the resin target aperture and background (see "Methods").

## A RSPT imaging setup (top view)

## B Target

## C Contrast ratio

## D Modulation Transfer Function

**Fig. 2 | Benchmarking of RSPT imaging instrument. A** Schematic representation of the RSPT setup (top view) showing the laser excitation (red) and Raman photon detection (carmine). The sample undergoes rotation and z displacement, with an inset showcasing the experimentally defined voxel dimensions. L lens, CL cylindrical lens, BP bandpass filter, LP longpass filter, AS aperture stop. **B** Schematic of the resolution target made of resin with apertures (1000 μm, 500 μm, and 400 μm). The target's midpoint was positioned at the cylindrical lens's focus, with a 45° incline relative to the optical system's perpendicular axis. **C** Plot indicating the relationship between the contrast ratio and the depth of field in RSPT imaging. **D** The calculated MTF for the RSPT system was based on the edge spread function from a projection of a cylindrical resin sample. From this, the line spread function (LSF) was derived and the MTF was computed by determining the magnitude of the Fourier Transform of the LSF. A demarcated cut-off at the 10% MTF threshold (1.35 cycles per mm) reveals the system's estimated resolution capability of 740 μm with its current magnification.

To estimate the RSPT system's overall optical performance in terms of spatial resolution and ability to reproduce varying levels of details, we measured projections of a cylindrical phantom and calculated the modulation transfer function (MTF) (Fig. 2D) (Supplementary Methods). The MTF illustrates the system's performance across a range of spatial frequencies. By examining specific points on the curve, such as the frequency at which the MTF 10% is 1.35 cycles per mm, we can estimate the system's limiting resolution to 740 μm. Considering our voxel size of 0.2 × 0.2 × 0.2 mm, the system meets the Nyquist sampling theorem for the determined resolution limit, ensuring that spatial details are captured and no undersampling occurs[38].

### Label-free 3D imaging of phantoms with molecular contrast

Supported by ray tracing simulations (Fig. 3A), we evaluated the capability of RSPT imaging to reproduce 3D printed semi-transparent resin samples with varying complex shapes (cylinder, cuboid, and a combined triangular prism/cylinder) (Fig. 3B). High quality Raman spectra could be acquired from resin phantom samples (signal-to-noise ratio (SNR) = ~52 at 900 cm$^{-1}$). A pure reference spectrum of resin (Fig. 3C) was measured for abundance estimation and reconstruction using non-negative least squares regression analysis. We calculated the sinograms showing all projections of a plane (0°–360°) and 3D rendering of the molecular abundance of resin. Tomographic

reconstruction allowed us to accurately replicate the phantoms in 2D and 3D based on Raman scattering (Fig. 3B and Supplementary Video 1–3). This data shows that RSPT can provide reliable 3D molecular imaging of complex transparent shapes.

We then evaluated the capability of RSPT imaging to reconstruct semi-transparent resin phantoms containing two profoundly different molecular components. We imaged a polyethylene terephthalate glycol (PETG) resin cylinder with an embedded polylactic acid (PLA) triangular-shaped sample (Fig. 3D). 3D reconstructions of the phantoms, based on the library Raman spectra of pure components, (Fig. 3C) demonstrate the capability of creating a label-free molecular contrast at a mesoscopic scale using RSPT imaging (Supplementary Video 4).

### RSPT imaging of ECM in native and living tissue-engineered constructs

We next investigated the utility of RSPT for performing non-destructive, label-free bioimaging of the spatial distribution of ECM in engineered tissues, albeit with lower spatial resolution than in phantoms due to optical scattering. Tissue engineering is a growing therapy for the treatment of degenerative pathologies (e.g., musculoskeletal, cardiovascular) that aims to regenerate functional tissues for clinical replacement of injured or diseased tissue. TE uses a variety of

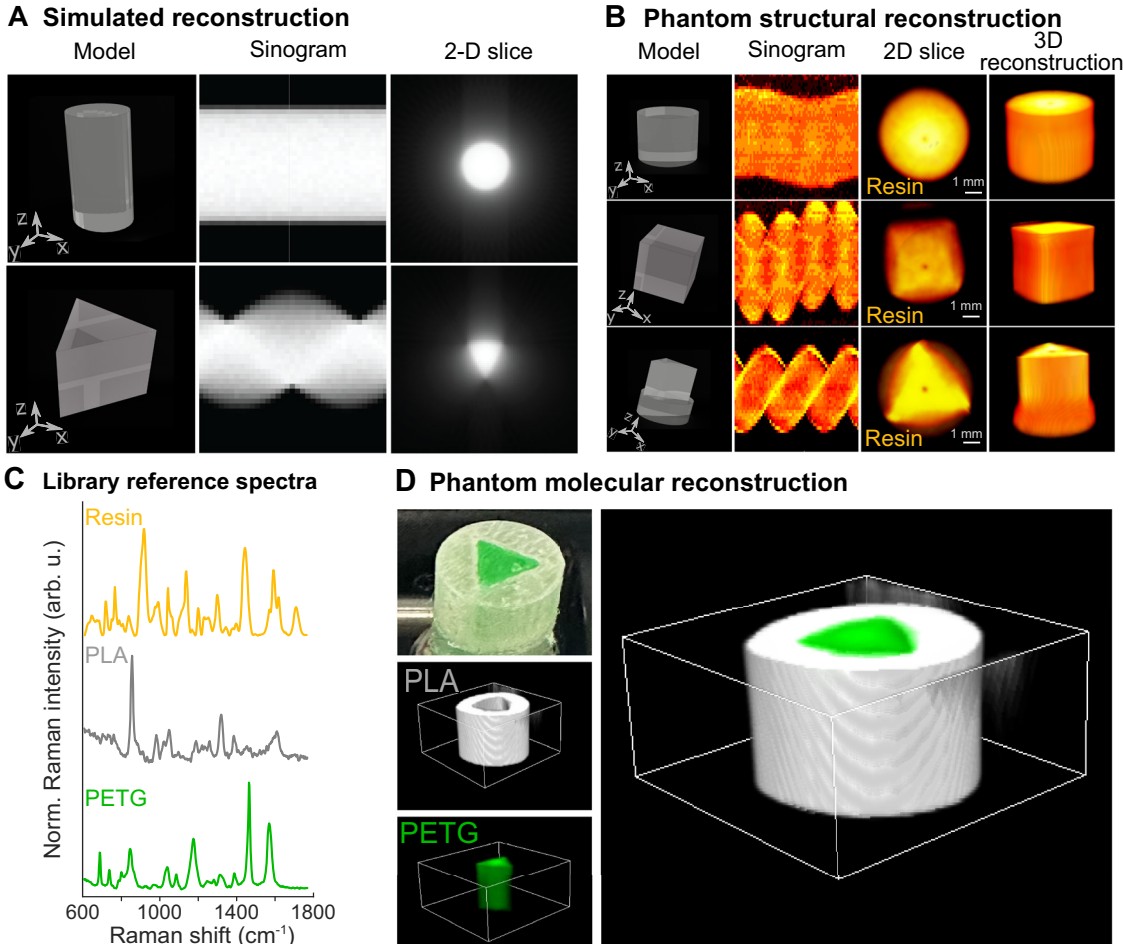

**Fig. 3 | Simulated and experimental RSPT imaging of phantoms. A** Simulated reconstruction using ray tracing depicts sinograms and 2D reconstructions of the cylinder and prism-shaped samples. **B** Imaging of resin phantoms with diverse geometries, including cylinder, cuboid, and a combination of prism and cylinder. Also shown are sinograms, 2D reconstructions of the basal slice of each sample, and the 3D rendering. **C** Pure library Raman component analysis of resin polymer, polylactic acid (PLA), and polyethylene terephthalate (PETG). **D** RSPT imaging and 3D reconstructions of a two-component sample: a hollow resin cylinder encasing a PETG prism. For all reconstructions, the background has been empirically removed with a threshold for visualization.

biomaterials, cells, and growth factors to generate repair tissues that recapitulate the composition, structure, and function of native tissue[39-41]. Given the influence of ECM organization on tissue functionality (e.g., mechanical load transmission)[39-41], non-destructive, label-free quantitative imaging of the spatial distribution, and heterogeneity of ECM may be critical for the clinical evaluation of the quality of engineered tissue grafts[42,43]. For cartilage TE, protocols aim to create neocartilage with an ECM composition that matches that of the native tissue, which is comprised of high levels of glycosaminoglycans (GAGs) interspersed within a type II collagen fibril network. It is this ECM network that lends the tissue its unique ability to provide lubricity and support the mechanical load.

Here, a cartilage tissue engineering model system is employed, whereby articular chondrocytes are seeded in a 2% *w/v* agarose hydrogel scaffold and cultivated to generate a neocartilage repair tissue. The technique strives to promote the cellular deposition of cartilaginous ECM constituents in order to recapitulate the composition of native cartilage, consisting of concentrated GAGs interspersed within a type II collagen fibril network. In this work, bovine chondrocyte-seeded agarose tissue constructs (Ø6 × 2.3 mm) were cultivated in a chondrogenic medium supplemented with transforming growth factor beta (TGF-β) for up to 56 days. Constructs of this size exhibit pronounced spatial ECM heterogeneities due to transport limitations of TGF-β and other nutrients[44]. The TMFP of native cartilage

is close to ~1 mm in the NIR range ~700 nm[45]. This will therefore naturally lead to image blurring for 7–8 mm thick samples. Endpoint RSPT imaging was performed on constructs at day 0 (SNR = ~22 at 1660 cm⁻¹), day 28 day (SNR = ~12 at 1660 cm⁻¹), day 56 (SNR = ~11 at 1660 cm⁻¹) (Figs. 4A and S2). Further, for reference, RSPT imaging was performed on native bovine cartilage tissues (SNR = ~28 at 1660 cm⁻¹) which exhibit natural ECM heterogeneities, whereby GAG and collagen concentrations increase as a function of depth from the articular surface. The library Raman spectra of pure reference molecules (GAG, type II collagen, agarose, and water) were measured for the multivariate regression analysis and 3D reconstruction (Fig. 4B and Supplementary Video 5–7) followed by 3D reconstructions of the molecular constituents (Figs. 4C, D and S3). Exemplar cross-sectional molecular images portraying tissue heterogeneities were also extracted from the principal x–z slices (Fig. S4). RSPT imaging of the constructs at day 0 showed high abundances of agarose and water and low abundances of collagen and GAG, consistent with freshly-cast constructs that have yet to synthesize cartilage ECM. As anticipated, there was some blurring and slight image degradation observed at the tissue edges, attributed to the transition between the tissue and background. By days 28 and 56 RSPT imaging demonstrates significant levels of GAG and collagen. In this model, ECM was distributed in a spatially dependent manner with constituents concentrated at the culture-media-exposed construct periphery (top-most region) but more dilute

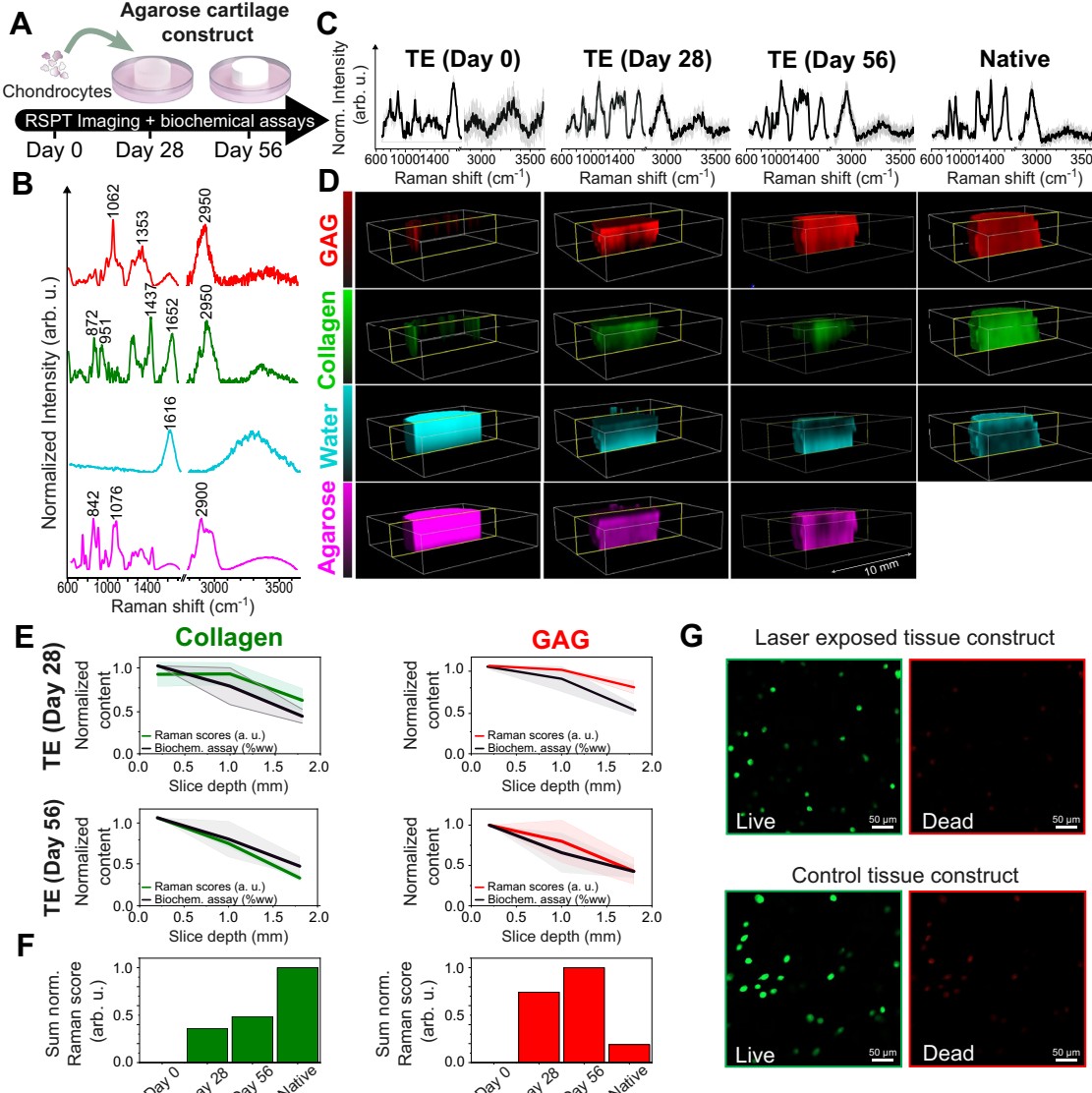

**Fig. 4 | RSPT imaging of tissue-engineered constructs and native cartilage tissues. A** Schematic of the experiment to create TE cartilage constructs, where chondrocytes were seeded in agarose and cultured in chondrogenic media for 56 days. **B** Normalized Raman spectra of library references components used in the 3D reconstruction (collagen, GAG, water, and agarose). **C** Representative mean Raman spectra ±1 standard deviation of various culturing time points, as well as for native cartilage tissues. **D** RSPT 3D rendering of the GAG, collagen, water, and agarose for different culturing time points, as well as for native cartilage. The color scale is arbitrary for display purposes (refer to **D** for quantitative analysis). For all reconstructions, the constant background signal has been removed with a threshold for visualization purposes. **E** Normalized mean sum of collagen (green) and GAG (red) RSPT-derived scores showing overall contents for day 0, day 28, day 56, and native samples. Data are presented as mean values ±1 standard deviation. **F** Normalized mean sum of collagen (green) and GAG (red) contents for day 0, day 28, day 56, and native samples, expressed as a proportion of the total sum of all components (GAG + collagen + water). **G** Live/dead assay fluorescence imaging conducted on living chondrocytes in a hydrogel. The imaging was conducted using calcein AM (green) and ethidium homodimer-1 (red) channels. Living cells emit green fluorescence, whereas deceased cells display red fluorescence (*n* = 2 replicates).

in the tissue center (bottom-most region), consistent with prior ex vivo characterizations of ECM heterogeneities that result from nutrient/growth factor transport gradients in TE constructs[44,46]. RSPT reconstructions further demonstrated ECM heterogeneities in native cartilage explants, whereby GAG and collagen increased as a function of depth from the articular surface, and the abundance of water decreased, consistent with their established spatial distribution through the depth-dependent zonal regions in native cartilage.

Correlation analysis of RSPT-derived ECM spatial gradients with biochemical assays (1,9-dimethyl methylene blue (DMMB) assay for GAG and Hydroxyproline assay for collagen) demonstrated good agreement (Fig. 4E). Collagen and GAG concentrations detected by RSPT imaging were highest at the surface and decreased deeper into the TE constructs. For native tissues, the gradients were opposite and

increased deeper in the tissues in agreement with previous reports using microscopy and histology[46]. Additionally, we semi-quantitatively assessed the quantities of GAG and collagen (Fig. 4F) at day 0, day 28, and day 56, and in native tissue. This assessment was performed by summing the RSPT-derived scores for collagen and GAG across the entire sample volume, followed by normalization to total molecular content. The results indicated a relative increase in GAG by day 56 that even surpassed that observed in native samples. In contrast, measurements of collagen content demonstrated a gradual increase towards the native levels. Hence, RSPT imaging represents a powerful tool for non-destructive bulk tissue gradient and heterogeneity molecular analytics. In a parallel tissue culturing experiment, we further investigated whether tissue constructs could be imaged in a live setting and the effect of continuous laser exposure. Live chondrocytes

were encapsulated in alginate hydrogels and divided into an exposed group and a control group. The exposed group was subject to imaging with the RSPT system (duration ~3 h). The control group was taken out of the petri dish for the same duration and was subject to the same hydration as the measured sample. Additionally, thermal imaging was performed during tissue imaging, which showed no detectable changes in the temperature of the tissue samples during laser exposure due to the highly hydrated conditions (Supplementary Fig. S5). Afterwards a live/dead cell viability assay was performed on both samples. The staining revealed that exposure to 2 W laser power had no clear effects on cell viability likely due to the efficient heat dissipation and absence of tissue chromophores (Fig. 4G). The percentage of live cells in the laser-exposed sample was 73%, while the control group exhibited a slightly lower cell viability at 62%.

## Discussion

Currently, there are limited tools available for mesoscale imaging that permit the direct visualization of the molecular composition of the ECM without necessitating labeling. Raman microscopy offers imaging of tissue sections with molecular contrast but has limited depth selectivity. Confocal Raman microscopy, on the other hand, provides remarkable volumetric resolution with diffraction-limited capabilities in cellular and organoid systems[20]. SORS can be used for depth-dependent analysis of tissues but does not possess imaging capabilities. The Raman spectral projection-based tomography system we developed here offers volumetric imaging of bulk samples at the mesoscale and therefore effectively complements these traditional approaches. We showcased a scenario where employing Raman spectroscopy, known for its advantages in live cell imaging and label-free molecular contrast, proves beneficial despite the challenges of reduced imaging resolution and blurring caused by diffuse photon propagation.

There are two main mechanisms enabling efficient image formation in the developed technique. The instrument uses slice illumination, and the projections of Raman spectra are acquired by a one-dimensional fiber array. This effectively rejects Raman photons scattered out of the imaging plane reducing lateral blurring. Secondly, although telecentric imaging and the use of an aperture stop to control the depth of field together result in a lower effective numerical aperture and fewer Raman photons being collected, it ensures that mostly forward scattered Raman photons are captured. Hence, RSPT employs a similar approach to OPT where the imaging lens acts as a collimator, reducing the collection of diffuse photons. It should be noted that normalization of the compounded Raman spectra projections and post-normalization of each voxel after back projection is of paramount importance in obtaining relative molecular gradients. Since each voxel is forced to have unit composition, attenuation (i.e., absorption or scattering) of the laser light through the sample does not introduce major changes in predicted molecular abundance levels, but rather mostly affects the signal-to-noise ratio.

We demonstrated the capability of RSPT imaging to volumetrically replicate the geometry of phantoms with molecular contrast. The minor artifacts in reconstructed shapes introduced during the image generation process can largely be attributed to refractive index mismatch. Immersing tissue samples in refractive index matching liquids to eliminate refractive mismatches might be feasible. The reconstruction of phantoms with different molecular components demonstrates that the compound Raman spectra can largely be deconvolved using multivariate regression analysis coupled with back projection of the molecular abundances.

We applied the technique in a TE setting, aiming to demonstrate its suitability as a tool to monitor the ECM composition and potential functionality of tissue replacements. Indeed, mimicking the native cartilage's ECM content and structural organization may be key in their long-term efficacy, but remains challenging[44]. Hence, the availability of

new semi-quantitative and non-destructive imaging tools such as RSPT can be vital for temporally assessing living engineered tissues. Native and engineered cartilage musculoskeletal connective tissues (e.g., cartilage, tendon, meniscus, intervertebral disk) represent ideal applications for RSPT due to their inherent avascularity and ECM-rich composition. The agarose scaffold is optically transparent, and the absence of vascularity and significant light absorption in native tissue enables relatively intense Raman signals to be generated and captured even through ~8–10 mm of bulk TE and native tissues. Therefore, the imaging results largely mirrored the ECM heterogeneities of engineered and native cartilage specimens, aligning with previous microscopic findings using confocal Raman microscopy of tissue sections, as well as biochemical assays[46]. The system also effectively captured images of live TE constructs containing chondrocytes within hydrogel matrices over 3 h without cell death. Hence, prolonged exposure (less than 180 min) to 2.0 W of 785 nm laser excitation does not jeopardize cell viability. Indeed, we did not observe any significant heating of the tissues, likely due to the efficient heat dissipation and absence of any significant tissue chromophores.

RSPT, despite its capabilities, is not without limitations. The imaging resolution for tissues depends on the sample type and size. It's important to note that the benchmarked (optical) imaging resolution relied on projections rather than a reconstructed 3D image. We consider this approach fair for this application as it helps avoid biases (e.g., filtering) introduced by reconstruction algorithms. The reconstruction of tissues requires prior knowledge of the key molecular makeup. The number of molecular constituents in the library can, however, be arbitrary and the quality of the image reconstruction will depend on their abundance and Raman scattering cross-section of each molecule. The computational framework assumes a linear relationship between molecular abundance and Raman signal intensity. Since the laser excitation has a Gaussian profile, absolute Raman signal quantification remains challenging highlighting the importance of normalization. There is considerable potential for further enhancing RSPT imaging. Here we used simple backpropagation reconstruction but more advanced reconstruction methods such as iterative reconstruction algorithms and scattering correction factors could potentially enhance the reconstruction[47]. Further, there are multiple avenues to explore for improving instrumentation including microscopic Raman tomography, incorporating resolution enhancement by direct imaging onto the spectrometer input slit, or even wavefront shaping approaches for better laser excitation control. Exploring the integration of multimodal OPT fluorescence imaging and RSPT presents exciting prospects for advancing functional imaging through the utilization of complementary contrasts.

The RSPT platform offers a new modality for label-free, volumetric mesoscale imaging of transparent and semi-transparent samples, achieving sub-millimeter optical spatial resolution with molecular contrast. Using molecular reconstruction of rich Raman spectral data, RSPT accurately reproduces transparent phantoms while also offering 3D molecular contrast. We also showed that there exists a favorable balance in which the benefits of Raman spectroscopy, such as its suitability for live cell imaging and label-free molecular contrast, outweigh the drawbacks of reduced imaging resolution and blurring caused by diffuse photon propagation enabling the temporal semi-quantification of molecular gradients within bulk tissues. These results underscore RSPT's potential as a platform for non-invasive molecular analysis, offering label-free tracking of molecular gradients and heterogeneity within living tissues.

## Methods

### RSPT instrumentation

The RSPT instrument (Fig. 1A) consists of a rotary stage (HDR50/M Thorlabs) to provide a full 360-degree rotation of the sample. The excitation and collection transmission paths were oriented at a small

angle (185°) to each other to prevent Raman photons from being generated inside the long-pass filters. For excitation, a custom wavelength stabilized 785 nm laser with a maximum output power of 2.0 W (Innovative Photonic Solutions) was fiber-coupled through a 365 µm low-OH multi-mode fiber and collimated using an aspheric condenser lens and then expanded using a Keplerian lens system ($f = 30$ and $f = 100$ mm, Thorlabs). A plano-convex round cylindrical lens ($f = 50$ mm, Thorlabs) was used to generate light sheet illumination in the sample plane. The laser light was passed through a 785 nm MaxLine laser clean-up filter (LL01-785-25, Semrock) to remove background signals. Samples were placed on the rotational stage and a motorized linear translation stage (LTS300/M, Thorlabs) allowing displacement in the z-direction. The generated Raman signals in a slice of the sample were imaged with a set of lenses ($f = 100$ mm, $f = 25$ mm, Thorlabs) directly onto a custom-made linear fiber array (43 fibers, 50 µm, Ibsen Photonics) acting as a spectrometer slit. A stop aperture was placed in between the lenses to ensure the collection of collimated light. A 785 nm EdgeBasic long-pass filter (BLP01-785-25, Semrock) was used on the detection side to remove the Rayleigh scattered light. The linear fiber bundle was directly coupled into a high-throughput Raman spectrometer (Eagle, Ibsen Photonics) covering both the fingerprint (600–1800 cm$^{-1}$) and high wavenumber range (2700–3600 cm$^{-1}$). The spectrometer was equipped with a back-illuminated deep-depletion CCD (Andor iVac, Oxford Instruments). Since tissue samples are susceptible to dehydration under the influence of laser exposure, we integrated an ultrasonic humidifier to prevent the samples from drying out over a 180-min scan and we continuously hydrated the sample with PBS.

## Instrument control software
We developed software for controlling the RSPT imaging instrument in the Python environment including a graphical user interface (GUI) (Fig. S1). The GUI was designed to record single projections, as well as a complete sequence of projections, where the integration time of the measurement, angle rotation resolution, and z-distance between the slices can be adjusted. Additionally, a raw CCD image and mean spectrum of the ongoing measurement are displayed for real-time visualization of the data quality during imaging.

## Preprocessing of CCD images
All spectral preprocessing and analyses of CCD images were performed within the MATLAB environment (Version 2020b, MathWorks Inc.) (Fig. 1B, C). Briefly, we first corrected spectrometer aberrations. A broadband light source spectrum was captured, providing registration points for polynomial transformation across the CCD. The derived spatial transformation was then applied to all the projection datasets. After aberration correction, the 43 imaged fibers were software binned on the CCD camera, with each fiber comprising five rows of pixels. The spectra underwent wavelength calibration using a Mercury–Argon source (HG-1, Ocean Insight) and cropped to the fingerprint wavenumber range spanning (600–1800 cm$^{-1}$) and high-wavenumber range (2700–3600 cm$^{-1}$). Cosmic rays were removed with a threshold and the cropped spectra were then smoothed utilizing a Savitzky-Golay algorithm (polynomial order = 2, window size = 5). Subsequently, a 3rd-order constrained polynomial fit was employed for effective elimination of autofluorescence in the fingerprint range. A 1st order polynomial was employed in the high wavenumber range. The Raman spectra were then normalized to their integrated areas in the fingerprint and high-wavenumber range, respectively. All projection data were analyzed together in a single standardized workflow for comparative analysis[48].

## 3D molecular image reconstruction
Molecular contrast from the Raman spectral projection imaging dataset was created by applying non-negative constrained least-squares using a library of reference spectra to calculate the abundance of each component in the compounded projected Raman spectra. Sinograms were constructed for each component by combining false color abundance images for every projection, resulting in one sinogram per library component. An inverse radon transformation back projection algorithm was then applied for the image reconstruction from each component sinogram. To ensure consistent value scaling of the abundances, post-vector normalization was applied to each voxel ensuring the imaging of relative gradients reducing the impact of laser attenuation. The workflow for RSPT image reconstruction including the CCD images and spectra is presented in Fig. S3. Data was interpolated prior to 3D rendering for visualization purposes that was performed in ICY version 2.5.2.0 and Autodesk Inventor.

## Ray tracing and simulations
The RSPT imaging instrument was simulated using Zemax OpticStudio 14.2 software. The simulation included two bulk samples with distinct geometries: a cylindrical and a triangular sample. To incorporate scattering, we employed a Mie scattering model[49]. The parameters used for Mie scattering, such as mean path, particle index (refractive index), size, density, minimum threshold, and transmission, were estimated based on the optical properties of materials utilized for 3D printing[50]. The fraction of incident light that passes through a material or sample, the transmission parameter was estimated using Lambert–Beer's law and the absorption coefficient[51]. Additionally, we developed a custom script that virtually rotates the sample and records the stimulated pixel intensity at the detector for each projection. To process the simulation results for 3D reconstruction, we employed a custom MATLAB script to generate a sinogram and reconstruct the image. This approach allowed us to model, analyze, and optimize the performance of the RSPT imaging system.

## Phantom samples development
Phantom samples in various shapes (cylinder, cuboid, and triangular prism) were designed with Autodesk Inventor software and then 3D printed with a UV resin printer (Elegoo Saturn 8k and Formlab Resin V2) using semi-transparent resin. The PLA samples were printed on an BambuLab A1 Mini, FDM 3D printer.

## Native cartilage tissue
Articular cartilage explants were procured from the femoral condyles of 3–6 months old bovine calves (Green Village Packing Co., Green Village, NJ, USA). Explants (n = 2) were trimmed to their final thickness by excising a portion of their deep zone, while the articular surface remained intact. Specimens were fixed in a 4% formaldehyde solution in DI water, supplemented with ethanol and acetic acid, for 24 h. Before imaging, the samples underwent three successive 1× phosphate-buffered saline (PBS) washes, with each wash lasting 24 h.

## Culturing of TE constructs
Articular cartilage was excised from eight bovine carpometacarpal joints (3–6 months old, Green Village Packing Co). Chondrocytes were isolated via tissue digestion in Type IV Collagenase (Worthington) at 37 °C for 17 h. Isolated chondrocytes were mixed in 2% w/v agarose (type VII, Sigma) at a density of $30 \times 10^6$ cells/mL to generate Ø6 × 2.3 mm tissue constructs. Constructs were cultured in chondrogenic media consisting of high glucose Dulbecco's Modified Eagle's Media (DMEM, Gibco) supplemented with 1 mM sodium pyruvate, 50 µg/mL L-proline, 100 nM dexamethasone, 1% ITS+ premix (Corning), 1% PS/AM antibiotic–antimycotic, and 50 µg/mL ascorbate-2-phosphate (Sigma) at 37 °C with regular media changes (three times per week) for a duration of 56 days. Human recombinant TGF-β3 (R&D Systems) was supplemented in media for the initial 14 days of culture. Samples (n = 2 per time point) were removed from culture for tissue

characterizations at days 0, 28, and 56 and subjected to the afore-mentioned fixation and washing protocol prior to RSPT imaging.

## Biochemical assay analysis
The spatial distribution of ECM content in engineered constructs was determined based on our established protocol[44]. Briefly, constructs were axially sub-cored with a 2 mm biopsy punch. Extracted cores were transversely cut into ~800 μm sections. Each section was weighed and digested in proteinase-K. Tissue digests were subjected to the DMMB assay for determination of GAG content. Digests were further sub-jected to acid hydrolysis and collagen content was determined via orthohydroxyproline assay. GAG and collagen contents were normal-ized to the sample weight to derive the absolute concentration in terms of ($\%w/w$)[52]. This analysis yields the concentration of each ECM component as a function of depth from the media exposed surface, allowing for comparison of biochemical content to RSPT imaging measures.

## Encapsulation of chondrocytes in alginate hydrogels
Alginate hydrogels were fabricated as described previsuly[53]. Briefly, sodium alginate (Sigma-Aldrich, 71238) was treated with 3 Mrad gamma irradiation to obtain low molecular weight alginate. Low molecular weight alginate was then dissolved and dialyzed (35 kDa cutoff) in Milli-Q water for three days. The final product was purified with activated charcoal, sterile filtered, frozen at −20 °C and lyophi-lized. Chondrocytes were isolated from calf femoral condyles and cultured in the growth medium containing hgDMEM (Thermofisher, 42430082), supplemented with 10% ($v/v$) fetal bovine serum, 1% ($v/v$) antibiotic–antimycotic (Thermofisher, 11570486), and 1% ($v/v$) non-essential amino acids (Thermofisher, 11140050). Cell encapsulation was performed as described previously[54]. Confluent chondrocytes at passage one were dissociated and resuspended in serum-free DMEM at a final concentration of 50 million cells mL$^{-1}$. Three percent ($w/v$) alginate in serum-free DMEM and containing cells were placed in a Luerlock syringe (VWR, LOCA120006IM). Four hundred eighty-eight mM calcium sulfate in serum-free DMEM was loaded into another Luerlock syringe and mixed with the cell-alginate solution through a Luerlock connector. The mixture was then deposited between hydro-phobic glass plates spaced 1 mm apart and allowed to gel for 45 min. Cell-laden hydrogels were cut into 6 × 6 mm squares and cultured in growth medium supplemented with 0.05 mg mL$^{-1}$ L-ascorbic acid-2-phosphate (Sigma) and 1 mM calcium chloride (Sigma) for 21 days.

## Cell viability assay
Live/Dead (Invitrogen, L3224) staining was performed according to the manufacturer's instructions. In brief, 500 μL cDPBS containing 1 μM calcium chloride and supplemented with 2 μM Calcein AM, 4 μM ethi-dium homodimer-1, and 1 μg mL$^{-1}$ Hoechst 33342 (Thermofisher, 62249) was added to chondrocyte-laden alginate hydrogels and was incubated for 1 h. Hydrogels were then washed twice in DPBS con-taining 1 μM calcium chloride and imaged on a laser scanning confocal microscope (Zeiss). Cell viability was assessed using ImageJ software. After segmenting and counting the stained cells in the acquired images (30 images per sample), the percentage of live cells was calculated by dividing the number of live cells by the total cell count (live plus dead cells).

## RSPT imaging of phantoms, TE constructs, and native cartilage
For these experiments, we used an FOV of 9 mm and a magnification of ~0.4 with a fully open stop aperture to increase collection efficacy. For phantoms, we acquired a full 360° scan with projections taken every 5° with an integration time of 4 s, and the z-step size was 500 μm. For tissues, we performed a 360° scan with projections taken every 15° with an integration time of 30 s per projection to accumulate enough Raman signal. Samples had diameters of ~8 mm and a height of ~3 mm

(TE constructs) and ~5 mm (native cartilage). The samples were hydrated intermittently with 5 μL of PBS between each slice measure-ment. Imaging of tissue-engineered cartilage presented more chal-lenges than native samples. Prior to RSPT imaging, a volume of 10 μL of PBS was introduced into the sample holder. This procedure ensures that the sample remains optimally hydrated. In addition, a humidifier containing deionized (DI) water was placed in the chamber to establish a humid environment. Post each measurement iteration of a specimen z-slice, an additional 10 μL of PBS was dispensed onto the sample, ensuring consistent hydration conditions throughout the measurements.

## Measurement of the library of purified molecular constituents of tissues
Pure components spectra of the native cartilage and TE constructs constituents, including collagen II (Sigma Aldrich), sulfated GAG (Sigma Aldrich) and agarose (Sigma Aldrich), and water were measured in the RSPT system with an integration time of 10 s.

## Reporting summary
Further information on research design is available in the Nature Portfolio Reporting Summary linked to this article.

## Data availability
The data that support the findings of this study are available from the corresponding author upon request.

## Code availability
The software used in this study is not publicly available due to com-mercial applications. However, requests for code access for academic research purposes can be directed to the corresponding author.

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

## Acknowledgements

We thank the National Centre for the Replacement, Refinement, and Reduction of Animals in Research (NC3Rs) for funding this work (NC/C018202/1 to M.A.B.H. and M.S.B.). This work was supported by GlaxoSmithKline and Galvani Bioelectronics, with co-funding from the Engineering and Physical Sciences Research Council (EPSRC). M.H.Y. acknowledges funding from the China Scholarship Council (CSC). E.G. is grateful for support from the EPSRC (EP/V04723X/1). We like to thank Tianbai Wang for assisting fabrication of TE cartilage constructs.

## Author contributions

E.S. and A.R.W. performed the experiments, interpreted the data, generated the figures, and wrote the manuscript. M.J., D.R.M., and M.H.Y. performed experiments and contributed to scientific discussion and data analysis. V.K. and S.V.P. performed experimental work and contributed to scientific discussion. E.G. contributed with scientific

discussion and data interpretation. M.B.A. contributed to scientific discussion, experiments, and data interpretation. M.S.B. and M.A.B.H. designed the study, interpreted and analyzed the data, and wrote the manuscript.

## Competing interests

M.S.B., M.A.B.H., S.V.P., and A.R.W. hold the following patent application (reference 20220160233). The remaining authors declare no competing interests.
