## [Peer Review File · Nature Communications]

Label-free 3D molecular imaging of living tissues using Raman Spectral Projection TomographyREVIEWER COMMENTS

Reviewer #1 (Remarks to the Author):

The paper reports the development of Raman imaging instrument capable of providing 3-D molecular contrast in transparent and semi-transparent samples. The paper is well written, the methodology is well described, and the results are convincing. I think it will advance the field, so I would recommend publishing it. The only suggestions are to perhaps move some of the supplementary material (figures S3-S5) into the main paper. Also, I can barely see the scale bar in figure 3, and it seems to be absent in figure 4, so it might be worth fixing it too.

Reviewer #2 (Remarks to the Author):

The authors report a label-free volumetric imaging tool, which is capable of providing 3D molecular contrast in transparent and semi-transparent samples with sub-millimeter resolution. They validated it in detecting molecular gradients and extracellular matrix heterogeneities in fixed and live tissue-engineered constructs and explanted native tissues. This work is somewhat interesting, but there are still issues to highlight in terms of experimental verification and innovation that prevented its publication in Nat Commun. Specific issues are as follows:

1. There have been several works to perform Raman volumetric imaging, for example Raman projection tomography, pulse-sheet chemical tomography, etc. The authors are suggested to elaborate and demonstrate what are the differences and advantages of this work over the published work. In particular, there is a need to validate its advancement.
2. In Introduction, "Finally, we demonstrated that the ..., highlighting the capability of the tool for temporal monitoring of TE...". How to realize the temporal monitoring? In this paper, the 42-day interval used to verify this concept (Fig.4) is a little reluctant, and the change of the intermediate time points of sample cannot be seen, which is not rigorous.
3. The Raman spectral projection tomography instrument was developed to collect the spectral signal of the complete samples by rotating and translating. This will greatly increase the data acquisition time, resulting in phototoxicity, etc. Is it suitable for a long-term imaging of live biological samples? Please add more details.
4. "Additionally, we semi-quantitatively assessed the composition of GAG and collagen

(Figure 4E)" There is no Figure 4E in the Fig.4.

5.It is mentioned that the system can achieve sub-millimeter resolution, what is the specific value? Please provide experimental measurements.

6."We evaluated the capability of RSPT imaging to reproduce 3-D printed semi-transparent resin samples with varying complex shapes..." Do samples of different shapes have an impact on image reconstruction? How to reconstruct an image from Raman spectrum, select a single peak or multiple? Please add more details.

7."The cropped spectra were then smoothed utilizing a Gaussian filter with $\sigma=2.41$ " please confirm the value of σ

8.The Modulation Transfer function(MTF) was ascertained using a 2-D reconstruction of a cylindrical sample. What is the cylindrical sample?

9.In Data pre-processing (Fig. 1B), please clarify the meaning of the image? the system RSPT collects the spectral data, why is raw images from 36 angles in 2nd step?

10.In Fig.2, is the geometry of RSPT accurately depicted? There is no 9 mm of the field of view.

11.The spatial resolution of the 3D image reconstruction in Fig.3B is too low, it is difficult to visualize the details and the shapes of different phantoms.

Reviewer #3 (Remarks to the Author):

The authors described a method for chemical imaging at mesoscale based on transmission Raman tomography. The authors demonstrated that for a sample as large as $\sim 1\text{cm}$, the spatial resolution was $\sim 600\mu\text{m}$, and chemical information can be collected to identify chemical contents. Compared to currently available methods, by sacrificing spatial resolution, this technique is unique for its ability to quickly acquire large volumetric imaging with molecular contents. However, a similar technique has been demonstrated, so the authors might need to clearly explain its novelty against previously reported approaches before publication.

1), A similar instrumental setup has been reported in another paper, as cited in reference #23. Another paper also reported similar Raman tomography by using transmission Raman coupled with fiber bundles. The data quality does seem to be superior in this paper, but it's

not very clear what the technological improvement described in this paper to the previous one. It's also not very clear what is the fundamental reason that this paper has better image quality.

2), there are other bond-selective tomographic methods that are also capable of volumetric chemical imaging [2][3]. The authors may also want to include these in their introduction section.

3), the authors claimed that the spatial resolution was $\sim 600\mu\text{m}$. For this method, is the resolution along the excitation line the same as the perpendicular direction? In other words, is the axial resolution the same as the lateral resolution?

4), for figure 3 E and F panels, the authors should also show the Raman spectra of the sample, in addition to the current reference spectra. The same should also be shown for figure 4 B and E. This will help the readers evaluate the quality of the spectra, and help readers understand whether content identification is possible.

5), for the samples in figure 3 and 4, what is the signal-to-noise ratio? Is it possible to identify more chemical species than the current two species? If so, what is the limitation? As tablet samples, a common application of transmission Raman, as well as biological samples, which is the focus of this paper, are most likely composed of more than two chemical components. So the authors should at least discuss the possibility of differentiating complex chemical compositions for a broader impact.

6), the authors did not explicitly explain why spectra of agarose and water (especially water since water is most likely the most abundant content in biological samples, and usually shows the most prominent peaks in Raman spectra of biological samples) were very weak. A detailed explanation might help readers to understand more of the technique. Also should be noted that although the authors mentioned that referencing spectra of agarose and water were acquired, but they were not shown in figure 4. (page 6 'The Raman spectra of pure reference molecules (GAG, type II collagen, agarose, and water) were measured for the multivariate regression analysis and 3-D reconstruction (Figure 4B)')

[1]. Jennifer-Lynn H. Demers, Francis W.L. Esmonde-White, Karen A. Esmonde-White, Michael D. Morris, and Brian W. Pogue, "Next-generation Raman tomography instrument for non-invasive in vivo bone imaging," *Biomed. Opt. Express* 6, 793-806 (2015)

[2]. Zhao, J., Matlock, A., Zhu, H. et al. Bond-selective intensity diffraction tomography. *Nat Commun* 13, 7767 (2022). <https://doi.org/10.1038/s41467-022-35329-8>

[3]. Chen, X., Zhang, C., Lin, P. et al. Volumetric chemical imaging by stimulated Raman projection microscopy and tomography. *Nat Commun* 8, 15117 (2017). <https://doi.org/10.1038/ncomms15117>

Reviewer #1 (Remarks to the Author):

The paper reports the development of Raman imaging instrument capable of providing 3-D molecular contrast in transparent and semi-transparent samples. The paper is well written, the methodology is well described, and the results are convincing. I think it will advance the field, so I would recommend publishing it.

The only suggestions are to perhaps move some of the supplementary material (figures S3-S5) into the main paper. Also, I can barely see the scale bar in figure 3, and it seems to be absent in figure 4, so it might be worth fixing it too.

We appreciate the reviewer's suggestion and have therefore integrated some of the supplementary materials into the main body of the paper that will serve to enhance the overall readability of the manuscript. We have incorporated Figures S3 and S5 into the main article, ensuring that the scale bars are clearly discernible in all figures throughout the manuscript. Due to the complexity and depth of information presented in Figure S4, we have made the decision to retain it as supplementary material (Figure S3). However, to still take this suggestion into consideration we have ensured that readers can grasp its content effectively by making significant enhancements to Figure 1B-C in the main text. These improvements aim to convey a similar message as Figure S4 in a more conceptual manner, thus facilitating a better understanding for the readers of the workflow of the data analysis.

Reviewer #2 (Remarks to the Author):

The authors report a label-free volumetric imaging tool, which is capable of providing 3D molecular contrast in transparent and semi-transparent samples with sub-millimeter resolution. They validated it in detecting molecular gradients and extracellular matrix heterogeneities in fixed and live tissue-engineered constructs and explanted native tissues. This work is somewhat interesting, but there are still issues to highlight in terms of experimental verification and innovation that prevented its publication in Nat Communications. Specific issues are as follows:

1. There have been several works to perform Raman volumetric imaging, for example Raman projection tomography, pulse-sheet chemical tomography, etc. The authors are suggested to elaborate and demonstrate what are the differences and advantages of this work over the published work. In particular, there is a need to validate its advancement.

We thank the reviewer for the comment so that we can clarify why our technique is entirely different. Diffuse Raman tomography has been developed which offers label-free volumetric analysis in 3-D. The reconstruction of images in diffuse Raman tomography involves inverse modeling. This means that simply collecting Raman spectra from different angles around the sample is not sufficient for generating a clear 3-D image. Instead, finite element methods are employed to reconstruct the spatial distribution of Raman-active molecules within the sample volume. Diffuse Raman tomography is therefore also often combined with other imaging modalities such as micro-CT. The spatial resolution was not stated explicitly in prior diffuse Raman tomography work, so a direct comparison is not possible. Our approach is based on true projection principles removing the need for spatial priors. We now exhaustively demonstrate this in both phantoms and tissues with much better imaging than our initial submission. Further, we completely outline the method for molecular reconstruction in great details (Figure 1). This is a key advancement in the field is therefore: (1) Development of an instrument that has large field of view, high molecular specificity (full fingerprint + high wavenumber Raman spectra) and living tissue compatibility. (2) Development of a data

analysis framework for molecular reconstruction and (3) Demonstration in a translational tissue engineering setting. Another projection technique was recently introduced (Wang et al Science Advances 2023) for imaging of zebra fish using a single band pass filter (i.e., single-band approach) with 532 nm excitation. This study did not apply spectroscopy and does therefore not offer specific molecular imaging. Consequently, the authors may have inaccurately designated their work as Raman when, in reality, they have measured autofluorescence. It was also therefore also disappointing to note that the authors didn't show a single Raman spectrum to support their work. Additionally, the use of 532 nm excitation penetrates less in tissues and raises concerns about potential photodamage to cellular components making it less suitable for live experiments. Our technique addresses all of these issues by utilizing true Raman spectroscopic imaging analysis with high molecular specificity using verified Raman signals and the use of NIR light to enable live tissue imaging (see Reviewer 2/comment 3). We have elaborated as follows:

Page 3: "The scattering of photons in tissue, which results in diffuse propagation, is frequently described by metrics such as the transport mean free path (TMFP). The scattering phenomenon in tissue poses a challenge to achieving high-fidelity optical projections. Consequently, diffuse Raman tomography has been developed which offers label-free volumetric analysis in 3-D.^{25, 26, 27} The reconstruction of images in diffuse Raman tomography involves inverse modeling. This means that simply collecting Raman spectra from different angles around the sample is not sufficient for generating a 3-D image. Instead, finite element methods are employed to reconstruct the spatial distribution of Raman-active molecules within the sample volume.^{25, 26, 27} Diffuse Raman tomography is therefore also often combined with other imaging modalities such as micro-CT.^{28, 29} Another recent development in projection tomography involves using a bandpass filter to capture isolated Raman scattering bands using visible excitation.³⁰ However, this approach falls short in utilizing the comprehensive molecular information present in full Raman spectra, might be obscured by autofluorescence, and has not been demonstrated to be compatible with applications involving living tissues."

We appreciate the reviewer's suggestions to discuss other approaches for optical tomography. We have therefore also now cited Demers et al Biomed. Opt. Express 2015 as part of our already incorporated discussion on diffuse tomography approaches. Further, we have included discussion of pulse-sheet chemical tomography, bond selective diffraction tomography and SRS tomography as follows:

Page 3: "Various methods for volumetric imaging utilizing SRS have also been developed, including stimulated Raman projection tomography.^{17, 31} These methodologies are primarily applicable to microscopic sized samples. A recent development, pulse-sheet chemical tomography, offers a new approach for imaging millimetre-sized samples with molecular contrast, however, with limited chemical specificity owing to its spectral resolution of 160 cm^{-1} .³² Advanced bond-selective diffraction tomography employing photothermal infrared techniques has also been demonstrated to enable rapid (~20 s) tomographic reconstruction with high resolution, albeit within a microscopic field of view.³³ Considering the strengths and limitations of current techniques, there remains a clear demand for the advancement of an optical imaging method that combines high molecular specificity, a broad mesoscale field of view, and the capacity to image living tissues without causing damage."

Additional references discussed and included:

- Jennifer-Lynn H. Demers, Francis W.L. Esmonde-White, Karen A. Esmonde-White, Michael D. Morris, and Brian W. Pogue, "Next-generation Raman tomography instrument for non-invasive in vivo bone imaging," *Biomed. Opt. Express* 6, 793-806 (2015)

Zhao, J., Matlock, A., Zhu, H. et al. Bond-selective intensity diffraction tomography. *Nat Commun* 13, 7767 (2022). <https://doi.org/10.1038/s41467-022-35329-8>

- Chi Yang, Yali Bi, Erli Cai, Yage Chen, Songlin Huang, Zhihong Zhang, and Ping Wang, "Pulse-sheet chemical tomography by counterpropagating stimulated Raman scattering," *Optica* 8, 396-401 (2021)
- Chen, X., Zhang, C., Lin, P. et al. Volumetric chemical imaging by stimulated Raman projection microscopy and tomography. *Nat Commun* 8, 15117 (2017). <https://doi.org/10.1038/ncomms15117>
- L. Gong, S. Lin, Z. Huang, Stimulated Raman Scattering Tomography Enables Label-Free Volumetric Deep Tissue Imaging. *Laser & Photonics Reviews* 2021, 15, 2100069

2. In Introduction, "Finally, we demonstrated that the ..., highlighting the capability of the tool for temporal monitoring of TE...". How to realize the temporal monitoring? In this paper, the 42-day interval used to verify this concept (Fig.4) is a little reluctant, and the change of the intermediate time points of sample cannot be seen, which is not rigorous.

We really appreciate this valuable feedback, which has prompted us to enhance the completeness of our study. We agree that the 0-day and 42-day interval used to verify this concept at first may not fully capture the dynamic changes of the samples over time. We have therefore now conducted additional tissue imaging experiments to address these limitations raised. Specifically, we have imaged tissues at day-0, day-28, and day-56 and we have visualized the gradients in a much more comprehensive manner to allow direct assessment of tissue gradients in 2D and 3D. These new experiments feature intermediate time points at shorter intervals, enabling a more detailed observation of the changes in the samples over time. We have incorporated these new results into Figure 4, allowing for a more comprehensive and rigorous evaluation of the temporal monitoring capability of our tool. Also, we have expanded on visual effects by including 7 rendered supplementary videos of both phantoms and tissues. We believe that by including these updated experimental results, we have addressed the issue of rigour in demonstrating the temporal monitoring capability of our tool. As an additional verification, we want to highlight that our tissue engineering gradient exactly replicate the gradients observed under Raman (microspectroscopy) and histology as previously published in our prior work (Albro et al NPJ Regenerative Medicine, 3,3 2018) serving as a further validation of the tomographic reconstruction.

3. The Raman spectral projection tomography instrument was developed to collect the spectral signal of the complete samples by rotating and translating. This will greatly increase the data acquisition time, resulting in phototoxicity, etc. Is it suitable for a long-term imaging of live biological samples? Please add more details.

We utilize 785 nm excitation with 2W laser power distributed over a large area resulting in a power density of $\sim 20 \text{ W/cm}^2$. 785 nm light will not be absorbed in these tissue that are made of collagen and GAG and water. We performed a test of cell viability after 3h exposure. Live chondrocytes were encapsulated in alginate hydrogels and divided into an exposed group and a control group. The exposed group was subject to imaging with the RSPT system (duration ~ 3 hours), the control group was taken out of the petri dish for the same duration and was subject to the same hydration as the measured sample. Afterwards a Live/Dead Cell Viability Assay was performed on both samples. The staining revealed that upon exposure to 2W laser power had negligible effects on cell viability. The percentage of live cells in the laser-exposed sample was 73%, while the control group exhibited a slightly lower cell viability at 62%. We have included these results in the main manuscript in Figure 4 (as also requested by Reviewer 1) so this will be clearer. In addition, to validate this we have performed additional experiments by measuring heating of the sample using an infra-red thermal camera showing no temperature rise during exposure of the laser over 3h. These results suggests that heat

diffuses fast in the highly hydrated samples, and we have included thermal images in supplementary information Figure S5.

4. “Additionally, we semi-quantitatively assessed the composition of GAG and collagen (Figure 4E)” There is no Figure 4E in the Fig.4.

We believe there must either be some inconsistency in the PDF that has been exported and provided to the reviewer or there has been a misunderstanding since Figure 4 in the submission did indeed have a E panel as well as a caption for it. Hopefully the new Figure will clarify.

5. It is mentioned that the system can achieve sub-millimeter resolution, what is the specific value? Please provide experimental measurements.

We appreciate the reviewer for bringing up this point, as it was not initially clear in our manuscript. After extensive research, we believe that the way we calculated resolution in the manuscript was not fair since it is compounded by the optical resolution and the reconstruction (i.e., algorithms). Because resolution can be biased by algorithmic filtering, we have now opted to only state the (optical) resolution.

Page 6 *“To estimate the RSPT system's overall optical performance in terms of spatial resolution and ability to reproduce varying levels of details, we measured projections of a cylindrical phantom and calculated the modulation transfer function (MTF) (Figure 2D). The MTF illustrates the system's performance across a range of spatial frequencies. By examining specific points on the curve, such as the frequency at which the MTF10 is 1.35 cycles per mm, we can estimate the system's effective resolution to 740 μm . Considering our voxel size of 0.2 x 0.2 x 0.2 mm, the system meets the Nyquist sampling theorem for the determined resolution limit, ensuring that spatial details are captured and no undersampling occurs.”*³⁹

Given that the resolution can be adjusted for specific applications based on the required field of view and will vary depending on the optical properties of the tissue, we believe that explicitly stating this in the abstract may not add significant value. The limiting resolution is determined by the fiber size, which is 50 μm which we have mentioned. Therefore, the term "sub-millimeter resolution" in the abstract broadly encompasses the instrument's capabilities by indicating the scale at which it operates.

6. “We evaluated the capability of RSPT imaging to reproduce 3-D printed semi-transparent resin samples with varying complex shapes...” Do samples of different shapes have an impact on image reconstruction?

The reviewer raises an important point. We have since submission of our manuscript improved the instrumentation further. We have implemented an aperture stop in a telecentric setup to improve the depth of field and more accurately reconstruct the phantoms. We have included these new results in Figure 3 which show massive improvement in imaging capability. There are still very minor artifacts due to refractive index mismatch stated as follows:

Page 10: *“ We demonstrated the capability of RSPT imaging to volumetrically replicate the geometry of phantoms with molecular contrast. The minor artifacts in reconstructed shapes introduced during the image generation process can largely be attributed to refractive index mismatch. Immersing tissue samples in refractive index matching liquids to eliminate refractive mismatches might be feasible.”*

How to reconstruct an image from Raman spectrum, select a single peak or multiple? Please add more details.

We appreciate that it was not clearly explained in our manuscript. In reconstructing images from Raman spectra, we employ a multivariate approach, which involves considering multiple

peaks rather than selecting a single one. This strategy enables us to take full advantage of all spectral features, thereby enhancing the accuracy and fidelity of image reconstruction. We have now comprehensively improved Figure 1B-C by explaining conceptually every single step in the preprocessing and data analysis. Further, we have explained it more extensively in the text as follows:

Page 5: *“To enable molecular imaging and reconstruction, we employ multivariate regression of a library of purified reference components (Figure 1C) as well as a background spectrum.³⁶ In this approach we measure Raman spectra of purified basis constituents and utilize non-negativity least squares regression analysis to quantify the relative abundance of each constituent. This implies that the composite Raman spectra observed result from the accumulation of all Raman signals generated within the sample. For highly transparent samples, considering that only a negligible portion of the incident laser light undergoes Raman scattering, the laser intensity remains mostly unchanged through samples, thus supporting the general assumption for image formation in projection tomography. Finally, 3-D image reconstruction is performed using inverse Radon transform - backprojection of the regression component abundances for each slice (Figure 1C).³⁷ The molecular abundances within the resulting voxels are ultimately normalized to one, facilitating the extraction of relative molecular gradients in samples. This implies that laser attenuation will have less effect on the reconstruction.”*

On top of this, we have now split the Methods section into **“Preprocessing of CCD images”** and **“3D molecular image reconstruction”** to make it clearer. Further in this revision we have significantly expanded the capability of the system and data analysis by including both the fingerprint (600-1800 cm^{-1}) and (2700-3600 cm^{-1}). This is a massive improvement since the fingerprint offers the molecular specificity while the high wavenumber range gives improved sensitivity to for instance tissue hydration.

7. “The cropped spectra were then smoothed utilizing a Gaussian filter with $\sigma=2.41$ ” please confirm the value of σ The (.41) represents a literature reference indicating there has been a formatting issue with the pdf export to the reviewer. Upon double-checking, we confirm that the correct value of σ utilized for smoothing the spectra was $\sigma=2$ as stated in the original manuscript. However, in the newest pipeline we did not use the Gaussian filter and instead opted to use a standard Savitzky-Golay filter (polynomial order=2, window size=5) as this represents a more commonly used approach in Raman spectroscopy and optimizing this parameter is beyond the scope of this work.

8. The Modulation Transfer function (MTF) was ascertained using a 2-D reconstruction of a cylindrical sample. What is the cylindrical sample?

We thank the reviewer for pointing this out as this was unclear in our manuscript. We have now clarified this:

Page 14 *“The Modulation Transfer Function (MTF) of the optical system was measured using a projections of a cylindrical sample of resin. We opted to assess the MTF on a projected data since there a no computational bias introduced by the reconstruction. We averaged the Edge Spread Function (ESF) from projections of a cylindrical phantom of resin. From this, the Line Spread Function (LSF) was derived by differentiating the ESF. The MTF was subsequently computed by evaluating the Fourier Transform of the LSF and determining its magnitude. The MTF was normalized by its peak value and then graphed against spatial frequency, providing an insight into the spatial resolution and overall image quality. A 5th order polynomial fitting was employed to represent the MTF, as showcased in Figure 2D, where the MTF was charted versus the spatial frequency (cycles per mm).”*

9. In Data pre-processing (Fig. 1B), please clarify the meaning of the image? the system RSPT collects the spectral data, why is raw images from 36 angles in 2nd step?

The reviewer has a great point, and we thank for raising this to our attention as it has allowed us to improve the Figure. The image in Figure 1B represents the number of vertically aligned Raman spectra acquired from the CCD. We have made this clearer now as follows

Page 4 *“We have developed a Raman spectral projection tomograph (RSPT) instrument based on a telecentric transmission Raman geometry and by rotating and z-translating of the sample (Figure 1A). In this design we excite the sample and collect a full z-slice of Raman spectra in the fingerprint (600-1800 cm^{-1}) and high-wavenumber range (2700-3600 cm^{-1}) at a time to offer complementary molecular information.*

Page 5 *“The line-to-line fiber array was coupled to a high-throughput spectrometer and imaged onto a 2000×256-pixel charge coupled device (CCD) for a z-slice spatial acquisition of Raman spectra (Figure 1A). Following the binning of pixels covering individual fibers on the CCD, this configuration provides snapshots of 43 Raman spectra across the FOV. In this way, a full projection dataset can be obtained by rotation and z translation of the sample. To accompany the instrument, we have developed a comprehensive controlling software for fully automated acquisition of data and analysis (Figure S1).”*

To further expand on this, we have added additional details in Figure 1A. Specifically we have added an image of the CCD readout to more facilitate an easier understanding. The number 36 was set arbitrary in this case for projections captured at 10 degrees intervals ($360/10$). We agree with the reviewer that this is not the best way to show this. Hence, we have now changed this in the Figure 1B to reflect an arbitrary number of projections n .

10. In Fig.2, is the geometry of RSPT accurately depicted? There is no 9 mm of the field of view.

This figure initially portrayed the optics conceptually (top view) rather than a real optical schematic as we relied on ray tracing for this. We have now expanded on this figure to reflect components in the setup including lenses, filters, aperture stops etc. In addition, we have more clearly marked where we define the field of view (FOV) with a more realistic grid. It should be noticed that the imaging setup is very flexible so the FOV can be changed depending on the sample size. We have ensured that this is clear:

Page 6: *“RSPT imaging is flexible and allows for the use of various magnifications, numerical apertures and, consequently, different spatial resolutions by modifying the collection imaging optics. Here we benchmarked the RSPT with a FOV of ~9 mm (Figure 2A) and a magnification factor of ~0.4 (voxel size of 0.2 x 0.2 x 0.2 mm).”*

11. The spatial resolution of the 3D image reconstruction in Fig.3B is too low, it is difficult to visualize the details and the shapes of different phantoms.

We conducted new phantom imaging using an optimized imaging configuration, wherein we expanded the depth of field by adjusting the aperture stop on the detection side in a telecentric design (Figure 1). This resulted in a major enhancement in the imaging reconstruction quality, which we have now showcased in Figure 3. In order to provide readers with a comprehensive understanding of the improved imaging quality, we have produced dedicated videos for each of the phantoms, which are included as supplementary information (Supplementary video 1-4). Notice, the stated resolution of 740 μm is calculated for projection data (not reconstruction data). Our developed technique exhibits a high degree of modularity, allowing it to be customized for various resolutions based on the size of the sample under examination. By changing the field of view (FOV) and depth of field, we can effectively balance collection efficiency and spatial resolution. This adaptability proves particularly advantageous for samples with weak Raman signals, such as tissues, where sacrificing spatial resolution enables more rapid data acquisition while still retaining valuable spectral information. We have expanded on this in the discussion:

Page 9: “There are two main mechanisms enabling efficient image formation in the developed technique. The instrument uses slice illumination, and the projections of Raman spectra are acquired by a one-dimensional fiber array. This effectively rejects Raman photons scattered out of the imaging plane reducing lateral blurring. Secondly, although the telecentric imaging and the use of an aperture stop to control the depth of field together results in lower effective numerical aperture and fewer Raman photons being collected, it ensures that mostly forward scattered Raman photons are captured. Hence, RSPT employs a similar approach to OPT where the imaging lens acts as a collimator, reducing the collection of diffuse photons. It should be noticed that normalization of the compounded Raman spectra projections and post normalization of each voxel after backprojection is of paramount importance in obtaining relative molecular gradients. Any attenuation (i.e., absorption or scattering) of the laser light through the sample will therefore merely have effect on the signal to noise ratio, since each voxel is forced to have unit composition.”

Reviewer #3 (Remarks to the Author):

The authors described a method for chemical imaging at mesoscale based on transmission Raman tomography. The authors demonstrated that for a sample as large as ~1cm, the spatial resolution was ~ 600µm, and chemical information can be collected to identify chemical contents. Compared to currently available methods, by sacrificing spatial resolution, this technique is unique for its ability to quickly acquire large volumetric imaging with molecular contents. However, a similar technique has been demonstrated, so the authors might need to clearly explain its novelty against previously reported approaches before publication. 1), A similar instrumental setup has been reported in another paper, as cited in reference #23. Another paper also reported similar Raman tomography by using transmission Raman coupled with fiber bundles. The data quality does seem to be superior in this paper, but it's not very clear what the technological improvement described in this paper to the previous one. It's also not very clear what is the fundamental reason that this paper has better image quality.

We appreciate that the reviewer is raising this point, so that we can clarify it in our manuscript, since the reported imaging technologies are very different. The reported Raman tomography technique (#23) was based on *diffuse tomography* which is fundamentally different from our approach using projections. In diffuse tomography a laser excitation is injected into the tissue and then Raman spectra are measured at different geometrical locations. This has been implemented with fiber bundles (now #25, #26 #27). The image recovery of these techniques relies fully on diffuse tomography. Diffuse tomography utilises a finite element model to solve a set of coupled diffusion equations, which predict the way the excitation fluence and the Raman emission fluence travel through the highly scattering media. Since diffuse optical image reconstruction is ill-posed and ill-conditioned, regularization is often used to stabilize the image recovery (#23). Our approach is ideally suited for transparent and semi-transparent samples and exhibits higher accuracy under these conditions (refer to Figure 3). We show that there exists a favorable balance wherein employing Raman spectroscopy, with its advantages in live cell imaging and label-free molecular contrast, outweighs the reduction in imaging resolution and blurring caused by diffuse photon propagation. More importantly, in our method we can tailor to different spatial resolutions and field of views by changing the imaging optics.

2), there are other bond-selective tomographic methods that are also capable of volumetric chemical imaging [2][3]. The authors may also want to include these in their introduction section.

We thank the reviewer for the input regarding other bond-selective tomographic methods capable of volumetric chemical imaging [2][3]. We have now incorporated information about these methods into the introduction section of our work as also requested by Reviewer 2:

Page 3: “Various methods for volumetric imaging utilizing SRS have also been developed, including stimulated Raman projection tomography.^{17,31} These methodologies are primarily applicable to microscopic sized samples. A recent development, pulse-sheet chemical tomography, offers a new approach for imaging mm-sized samples with molecular contrast but with limited chemical specificity owing to its spectral resolution of 160 cm^{-1} .³² Advanced bond-selective diffraction tomography employing photothermal infrared techniques has also been demonstrated to enable rapid ($\sim 20\text{ s}$) tomographic reconstruction with high resolution, albeit within a microscopic field of view.³³ Considering the strengths and limitations of current techniques, there remains a clear demand for the advancement of an optical imaging method that combines high molecular specificity, a broad mesoscale field of view, and the capability to work effectively with living tissue.”

3), the authors claimed that the spatial resolution was $\sim 600\mu\text{m}$. For this method, is the resolution along the excitation line the same as the perpendicular direction? In other words, is the axial resolution the same as the lateral resolution?

The resolution in the horizontal and vertical direction is ultimately limited by the diameter of the fiber and sample and the choice of magnification in the imaging optics and therefore the field of view (FOV). This can be modified for various applications simply by changing the imaging lenses. After extensive research, we believe that the way we calculated resolution in the manuscript was not fair since it is compounded by the optical resolution and the reconstruction (i.e., algorithms). This is not fair since because algorithms such as filtered backprojection can inflate the resolution by filtering. We have now opted to only state the optical resolution ($740\ \mu\text{m}$).

Page 6 “To estimate the RSPT system's overall optical performance in terms of spatial resolution and ability to reproduce varying levels of details, we measured projections of a cylindrical phantom and calculated the modulation transfer function (MTF) (Figure 2D). The MTF illustrates the system's performance across a range of spatial frequencies. By examining specific points on the curve, such as the frequency at which the MTF10 is 1.35 cycles per mm, we can estimate the system's effective resolution to $740\ \mu\text{m}$. Considering our voxel size of $0.2\ \text{x}\ 0.2\ \text{x}\ 0.2\ \text{mm}$, the system meets the Nyquist sampling theorem for the determined resolution limit, ensuring that spatial details are captured and no undersampling occurs.³⁹”

The lateral resolution might be slightly better for tissue samples since we measure a single slice at a time. Measuring a single slice at a time reduces the collection of out-of-plane diffuse Raman photon contributions but this will largely be sample dependent and we have mentioned this:

page 4: “Measuring a single slice at a time reduces the collection of out-of-plane diffuse Raman photon contributions”

4), for figure 3 E and F panels, the authors should also show the Raman spectra of the sample, in addition to the current reference spectra. The same should also be shown for figure 4 B and E. This will help the readers evaluate the quality of the spectra, and help readers understand whether content identification is possible.

We thank the reviewer for these great suggestions and we agree. We conducted additional phantom imaging using an optimized imaging configuration, wherein we expanded the depth of field by adjusting the aperture on the detection side. This optimization resulted in a notable enhancement in the imaging reconstruction quality, which we have showcased in Figure 3. Furthermore, in order to provide readers with a comprehensive understanding of the improved imaging quality, we have produced 3D videos for each of the phantoms, which are included as supplementary information. We have included the Raman spectra in Figure 3 as well. Further in Figure 4 we now show all the library reference Raman spectra used in the regression analysis as well as well as representative examples of tissue spectra.

5), for the samples in figure 3 and 4, what is the signal-to-noise ratio? Is it possible to identify more chemical species than the current two species? If so, what is the limitation? As tablet samples, a common application of transmission Raman, as well as biological samples, which is the focus of this paper, are most likely composed of more than two chemical components. So the authors should at least discuss the possibility of differentiating complex chemical compositions for a broader impact.

We calculated the signal to noise ratio for the resin and tissues and we have now mentioned this in the results as follows:

“Supported by ray tracing simulations (Figure 3A), we evaluated the capability of RSPT imaging to reproduce 3-D printed semi-transparent resin samples with varying complex shapes (cylinder, cuboid, and triangular prism) (Figure 3B). High quality Raman spectra could be acquired simultaneously across the CCD (SNR=52 at 900 cm⁻¹).”

Page 8 *“Endpoint RSPT imaging was performed on constructs at day 0 (SNR=~22 at 1660 cm⁻¹), day 28 day (SNR=~12 at 1660 cm⁻¹), day 56 (SNR=~11 at 1660 cm⁻¹) (Figure 4A, Figure S2). Further for reference, RSPT imaging was performed on native bovine cartilage tissues (SNR=~28 at 1660 cm⁻¹) which exhibit natural ECM heterogeneities, whereby GAG and collagen concentrations increase as a function of depth from the articular surface.”*

It is certainly possible to estimate more chemical compounds in samples. Non-negativity least squares regression of an arbitrary number of compounds can be used to estimate these and this will fully depend on the sample/chemical in question. The fact that we are recovering the Raman signal in a projection tomographic setting will not have influence on this data analysis. We have therefore now explicitly addressed this question in the discussion as follows:

Page 11 *“The reconstruction will largely be sample dependent and requires prior knowledge of the key molecular makeup, but the number of molecular constituents can be arbitrary. By including all known constituents, this type of multivariate regression analysis generally will enable highly accurate reconstruction.”*

6), the authors did not explicitly explain why spectra of agarose and water (especially water since water is most likely the most abundant content in biological samples, and usually shows the most prominent peaks in Raman spectra of biological samples) were very weak. A detailed explanation might help readers to understand more of the technique. Also should be noted that although the authors mentioned that referencing spectra of agarose and water were acquired, but they were not shown in figure 4. (page 6 ‘The Raman spectra of pure reference molecules (GAG, type II collagen, agarose, and water) were measured for the multivariate regression analysis and 3-D reconstruction (Figure 4B)’)

It is indeed pertinent to note that while water is abundantly present in tissues, its molecular fingerprint within the 600 - 1800 cm⁻¹ range exhibits relatively weak signals. We therefore now included the high wavenumber range (2700-3600 cm⁻¹) as well. It is important to note that despite constituting approximately 90% of tissue composition, the contribution of water signatures to the overall spectra in the fingerprint range is minimal. In response to the reviewer, we have made significant improvements to Figure 4. We have expanded our analysis by including additional timepoints for the tissue imaging. Furthermore, in order to elucidate the spectral characteristics more comprehensively, we have not only incorporated agarose and water peaks but also provided representative examples of mean compound tissue Raman spectra ± 1 standard deviation. These additions serve to offer a more nuanced appreciation of the quality of the spectroscopic data within the context of the tissue analysis.

REVIEWERS' COMMENTS

Reviewer #1 (Remarks to the Author):

The authors have appropriately addressed the issues raised in the initial review and the paper now is ready for publication.

Reviewer #2 (Remarks to the Author):

The authors have addressed most of my previous concerns. However, this reviewer still not see the SIGNIFICANCE of this manuscript compared to the published work. The authors elaborate that this work could cover a wider Raman spectrum, but this is not a fatal SIGNIFICANCE for the existing work. The projection tomography-based 3D Raman techniques in the published work can achieve this by switching to a hyperspectral camera at the detector end or by using tunable filters with high spectral resolution. The large-scale samples emphasized by the authors are at the expense of spatial resolution, and again projection tomography techniques can be implemented by similar ideas. More importantly, the authors need to validate the advantages of the technique on large-scale, alive bio-samples (like embryo, zebrafish), i.e., what the technique can achieve that the published work cannot.

The authors should elaborate and demonstrate the groundbreaking contribution of the work done in this paper to the field of 3D Raman to be worthy of the high-quality requirements of Nat Comm.

Reviewer #3 (Remarks to the Author):

The authors have addressed my comments fully, I recommend publication for the current version.